# Parallel Acquisition of Plasma Membrane Ultrastructure and Cytosolic Protein Localisation in Cultured Cells via Correlated Immunogold SEM

**DOI:** 10.3390/cells9061329

**Published:** 2020-05-26

**Authors:** Isabell Begemann, Ulrike Keller, Harald Nüsse, Jürgen Klingauf, Milos Galic

**Affiliations:** 1Institute of Medical Physics and Biophysics, University of Muenster, Robert-Koch-Str. 31, 48149 Münster, Germany; isabell.begemann@uni-muenster.de (I.B.); kelleru@uni-muenster.de (U.K.); nusse@uni-muenster.de (H.N.); klingauf@uni-muenster.de (J.K.); 2Interfaculty Centre ‘Cells in Motion’, University of Muenster, Waldeyerstr. 15, 48149 Münster, Germany

**Keywords:** scanning electron microscopy, SE/BSE, CLEM, immunogold labelling, curvature-sensitive protein, BAR domain

## Abstract

Scanning electron microscopy (SEM) takes advantage of distinct detectors to visualise secondary and back-scattering electrons. Here, we report an integrated approach that relies on these two detection methods to simultaneously acquire correlated information on plasma membrane topography and curvature-sensitive cytosolic protein localization in intact cell samples. We further provide detailed preparation and staining protocols, as well as a thorough example-based discussion for imaging optimisation. Collectively, the presented method enables rapid and precise analysis of cytosolic proteins adjacent to cellular membranes with a resolution of ~100 nm, without time-consuming preparations or errors induced by sequential visualisation present in fluorescence-based correlative approaches.

## 1. Introduction

Recent advancements in microscopy techniques have opened up novel inroads to investigate fundamental biomedical processes at high spatial and temporal resolutions. These approaches generally rely on fluorescence- or electron-based microscopy techniques. Fluorescence microscopy is well-suited to study a wide range of protein properties, while details on specimen ultrastructure (e.g., surface topography) cannot be displayed, due to poor spatial resolution of fluorescence probes. Electron microscopy (EM), in contrast, allows detailed acquisition of a specimen with a resolution of <1 nm, but lacks information on protein localisation. To reliably extract ultrastructural information and compare it to that of proteins enriched at such sites, hybrid approaches that have correlated light and electron microscopy (CLEM) have proven useful [1,2,3]. Here, information from fluorescence-based images and electron micrographs are separately acquired and subsequently merged. However, despite significant improvements [4], final image alignment still yields substantial inaccuracies. These alignment errors originate not only from fluorescence-based limitations in axial and lateral resolution, but also from changes in shape and volume introduced during fixation, sample preparation, and transfer between microscopes. Even with a perfect overlap of the respective imaging areas, any specimen movements that may have occurred between acquisition on the fluorescence and electron microscope will yield alignment errors.

In scanning electron microscopy (SEM), surface features are generally detected using secondary electrons (SE). Electron microscopes are further equipped with a back-scatter electron (BSE) detector, whereby back-scattered electrons scale with, among other things, the atomic number of the element. Hence, by using gold-conjugated secondary antibodies, a protein of interest can be localised using the BSE detector, while the cellular surface can be simultaneously pictured via the SE detector. Not surprisingly, combined SE/BSE imaging analogous to CLEM has been frequently used for the parallel acquisition of protein localisation and cellular ultrastructures. These published approaches, however, take advantage of readily accessible epitopes at the cell surface [5,6,7]. To study intracellular structures, alternative methods have been developed that rely on subcellular fractionation [8], cryofracture [9,10], or chemically dissolved membranes [11] to expose interior regions. However, such approaches are not well-suited to study proteins attached to the plasma membrane. Likewise, cellular unroofing via mechanical means, stripping techniques, or strong shearing forces will randomly remove an unknown amount of the cell surface, thereby destroying the membrane interface and eradicating the proteins along with it [12]. Hence, to date no systematic characterization and optimization of SE/BSE has been accomplished to correlate plasma membrane features with intracellular protein targets in intact cultured cells.

Richards and Gwynn paved the way towards this goal [13]. In their studies, they successfully used BSE to display intracellular structures that were stained with heavy metals within an unstained resin. Similarly, the localisation of intracellular proteins was accomplished via BSE imaging of gold-conjugated antibodies [14]. Building on these studies, we set out to develop a method suitable to detect protein localisation at the cell interior and correlate it with ultrastructural cell surface information. Curvature-dependent signalling at cellular membranes plays a pivotal role in cellular processes, such as exo/endocytosis, filopodia formation, and cell migration [15,16,17,18]. Here, we took advantage of curvature-sensitive cytosolic proteins, which selectively enrich at transiently deformed plasma membrane sections, to establish a protocol for the parallel acquisition of membrane ultrastructure and cytosolic protein localisation in cultured cells via SE/BSE. Specifically, we optimised protocols from ours and other labs [1,13,14,19,20] for fixation, sample preparation, labelling, coating, and characterised labelling and detection accuracy for intracellular localisation in intact cultured cell samples. Altogether, this provides a versatile and simple method for correlative ultrastructure protein studies adjacent to cellular membranes.

## 2. Material and Methods

The method is structured into five work packages: preparation, fixation, immunolabelling, EM preparation (dehydration and coating), and imaging (Figure 1).

### 2.1. Holder and Sample Preparation

Holder: suitable glass object holders were cut into pieces—in our case, 3 × 7 mm. For later identification and orientation, the holders were marked from the bottom using a diamond marker. Holders were washed in acetone, absolute ethanol, and water for 5 min each. Following air-drying, sample holders were exposed for 15 min to UV light to ensure contamination-free cell substrates.

Sample preparation: depending on the cell type, substrate coating can be accomplished at this point. In the presented case, mouse fibroblasts (Leibniz Institute DSMZ, ACC-59) were plated on the object holders by applying a drop of suspended cells (~10 µL) in DMEM containing 4.5g/L D-glucose, GlutaMax-I and pyruvate (Gibco, 31966-021), 10% fetal bovine serum (FBS) (Biochrom AG, L11-004), and 1% penicillin/streptomycin (10,000 U/mL–10,000 µg/mL) (Biochrom AG, A2212). After letting cells attach for 30 min, the well was carefully filled with pre-warmed medium. Depending on the experimental design, cells at this point could either be directly fixed for later staining with antibodies directed against endogenous proteins (see Figure 2), or transfected with a fluorescently labelled protein of interest (see Figure 3). In the latter case, cells were transiently transfected using Lipofectamine 2000 (Life Technologies, 11668-027), according to the manufacturer’s protocol.

### 2.2. Fixation

Thorough fixation and preservation is essential for keeping proteins of interest attached to the membrane, as membrane tension [21,22], membrane lipid composition [23,24], and other factors may change its binding affinity [22]. Glutaraldehyde preserves good ultrastructure, but is a slow fixative and can deteriorate epitope-binding of the antibody. Similarly, osmium tetroxide, while an excellent stain for lipids, is not recommended, as antigenicity could be affected [25]. To secure proper fixation, cells were first dipped three times in PBS (Gibco, 10010-023) with 4% sucrose (Sigma, S7903), and subsequently fixed using 6% para-formaldehyde (PFA) (Ted Pella, 18505) in PBS containing 4% sucrose for 20 min at room temperature (RT).

### 2.3. Immunogold Labelling

In order to remove residual PFA, samples were washed three times for 5 min with PBS. To quench remaining free aldehyde groups, samples were incubated for 20 min with 100 mM NH_4_Cl (Carl Roth, K298.2) in PBS, followed by washing twice for 5 min with 2.5% bovine serum albumin (BSA) (Sigma, A9085.25G) in PBS. Permeabilisation was performed using 2.5% BSA and 0.1% Triton-X-100 (Sigma, T9284) in PBS three times each for 5 min. Primary antibody incubation was accomplished using anti-GFP (Abcam, ab6556) or anti-actin (Abcam, ab14128) diluted in PBS containing 2.5% BSA. In our case, the appropriate dilutions were 1:50 (anti-GFP) and 1:100 (anti-actin), well below the manufacturer’s recommendation. The equipped sample holders were dipped upside-down into a 20 µL droplet of diluted antibody on parafilm, and incubated for 6 h in a humidified chamber in the dark. After primary antibody incubation, cells were washed three times for 5 min in PBS containing 2.5% BSA to remove unbound primary antibody fractions, followed by washing them three times for 5 min in Tris-NaCl containing 2.5% BSA. A second permeabilisation was accomplished in Tris-NaCl containing 2.5% BSA and 0.1% Triton-X-100 three times, each for 5 min. Secondary gold-conjugated antibodies, at 10 nm (Abcam, ab27241) and 20 nm (Abcam, ab27242), were applied in a dilution of 1:5 to 1:10 for 4–5 h as before (i.e., parafilm, humified chamber, and in the dark). To remove unbound secondary antibodies, cells were washed three times for 5 min with Tris-NaCl containing 2.5% BSA. This was followed by three times washing for 5 min with Tris-NaCl buffer to remove excess BSA. In order to further stabilise antibody binding, as well as protein and cellular composition, samples were post-fixed with PBS containing 2.5% glutaraldehyde (GA) (Agar Scientific, R1011) for 1 h at RT. Finally, excess glutaraldehyde was removed by washing samples three times for 5 min with PBS.

### 2.4. Dehydration and Coating

To prepare the probes for scanning electron microscopy, an ascending alcohol row (30%, 50%, twice 70%, 90%, twice absolute ethanol) was applied for sample dehydration, each for 10 min. Next, the dehydrated samples were subject to critical point drying with liquid CO_2_. The samples were then mounted on aluminium specimen holders utilising Leit-C (Plano, G3300), an electronically conducting carbon glue. After 3 h of air-drying, the mounted samples were rotary-shadowed at RT with a 2.5 nm layer of chromium, using an angle of 65° and constant rotation (BAF 300, Balzer’s Union). Note that chromium oxidises rapidly. As thin metal coatings do not interfere with the BSE signal [26,27], a 1.2 nm protective layer of platinum carbon can be applied for long-term sample usage (optional).

### 2.5. Imaging

The prepared samples were investigated in a high vacuum using a high-resolution field emission scanning electron microscope (“in-lens” type, model S-5000, Hitachi Ltd.). Electron micrographs were captured using accelerating voltages between 6 kV and 30 kV. For the detection of gold particles, which were bound to the respective antibodies, the back-scattered electrons were collected utilising the BSE detector.

Confocal images were acquired with an EMCCD camera (Andor, DU888 Ultra), mounted on an inverted Nikon microscope (Nikon, Ti Eclipse RCD) equipped with a spinning disc unit (Yokagawa, CSU-X1). Stimulated emission–depletion (STED) images were acquired using a STEDYcon mounted on an inverted Nikon microscope equipped with a pulsed diode laser at 640 nm (Abberior Instruments, 1.2 mW), a 775 nm laser (1.2 W), a QUAD Scanner (90 µm × 80 µm (100×/1.4 NA oil), a single bandpass (650 nm–700 nm) filter, and a single photon-counting avalanche photodiode (APD).

## 3. Results

### 3.1. Detection Depth of Back-Scattered Electrons

Theoretical calculations suggest a possible electron beam penetration of ~2 µm through carbon (depending on the accelerating voltage) [28,29]. In order to experimentally determine the detection depth of electrons back-scattered from gold particles within the cell, we used carbon as a biomimetic substrate. In the first step, we deposited 10 nm and 20 nm gold particles on a carbon-coated EM grid, and measured the relative signal intensities at accelerating voltages of 6 kV, 10 kV, 20 kV, and 30 kV, respectively. Next, additional layers of carbon were deposited, from 24 nm to 238 nm. For 10 nm gold particles, we found a maximal detection depth of 44 nm at 6 kV, 88 nm at 10 kV, and 160 nm for 20 kV and 30 kV (Figure 2a and Appendix A). Depending on acceleration voltage, 20 nm gold particles could be resolved in depths of 88 nm (6 kV) to 238 nm (30 kV) (Figure 2b). Additionally, we determined the scattering area of the back-scattered electrons (i.e., the “halo”) in relation to applied voltage, gold particle size, and carbon layer thickness for each condition (Figure 2a,b). To that end, BSE images of 10 nm and 20 nm particles were automatically thresholded in ImageJ, using a variant of the isodata algorithm [30], and the area of the “halo” was plotted. Spherical blurring (i.e., the “halo” area) increased with carbon thickness and reduced acceleration voltage, providing an additional variable that can be used for determining the depth of individual gold particles underneath the carbon layer.

### 3.2. Validation of Localization Depth

The lamellipodium at the leading edge is a central part of the cellular migration machinery. With a thickness of ~100 nm to 200 nm along the *z*-axis, it is well in the range of the estimated detection depth of the presented method (Figure 2a,b). As a gold particle size of 20 nm was appropriate for the thickness of the cellular structure, and at the same time yields good labelling results, we continued the validation with this size. For the coating, we used chromium, due to its small granularity, high SE yield, and low amount of BSEs [27]. To determine whether gold-conjugated antibody labelling is specific in cellular samples, a positive control containing an antibody directed against actin, a well-established cytoskeletal marker [3], was imaged in SE and BSE modes. Specifically, cells were partially unroofed after critical point drying, and immunogold particles directed against actin were compared in unroofed and adjacent intact membrane areas. No statistically significant difference was observed (Figure 2c and Appendix A), arguing that membranes do not mask gold signals directed against cortical actin.

To further investigate the possibility of unspecific trapping of gold-conjugated antibodies underneath the cellular membrane, samples were partially unroofed and incubated in the absence of the primary antibody (Appendix A). Statistical analysis revealed no significant difference in gold particle count per area (Appendix A, graph). Hence, membrane or membrane-close structures do not seem to trap or mask unspecific gold particles. Collectively, these results establish that accurate and proper gold particle localisation can be achieved, excluding the possibility of obscuring or decreasing the BSE signal or unspecific trapping of 20 nm immunogold particles within cells.

Notably, and consistent with our carbon layering data (Figure 2a,b), we further observe spherical blurring (i.e., the “halo”) with gold particles directed against actin (Figure 2d). Pending proper calibration, ideally combined by deconvolution [31], these measurements may be suitable to not only determine lateral but also axial positioning of the probe in the cellular sample.

### 3.3. Applications

Finally, we next aimed to apply the method to biological systems. As the presented method is optimised for cellular systems not thicker than a few hundred nanometres (Figure 2a,b), or for the proteins being present adjacent to the plasma membrane, we probed the localisation of the curvature-sensing I-BAR domain of BAIAP2 that preferentially binds outward (i.e., convex) membrane deformations [15,16,17,18]. Confocal [32] and super-resolution [33] microscopy data showed enrichment of a fluorescently labelled I-BAR domain of BAIAP2 (I-BAR) over a cytosolic reference at the lamellipodial edge (Figure 3a). However, both approaches lack information on membrane topography. In contrast, correlated protein ultrastructure localisation via SE/BSE showed I-BAR protein enrichment selectively in highly curved membrane sections that form at the outermost rim of the lamellipodium (Figure 3b).

To further probe the validity of the method, we took advantage of filopodia, a sensory and migratory structure present in several cell types. The size of these subcellular structures is well within the detection range of SE/BSE microscopy and should host accumulated amounts of I-BAR proteins. Indeed, ratiometric and super-resolution images showed an enrichment of transfected I-BAR domain over a cytosol reference in filopodia (Figure 3c). Consistently, SE/BSE showed I-BAR enrichment in filopodial structures (Figure 3d).

As above (Figure 2a,b,d), we observe spherical blurring (i.e., a “halo”), with gold particles directed against the curvature-sensitive I-BAR domain (Figure 3e).

## 4. Discussion

### 4.1. Comparison with Existing Methods

CLEM is a powerful tool to probe for form–function relation in cells. However, such approaches classically rely on two imaging sessions that are separated by several preservation and processing steps, which all can yield alignment errors. Furthermore, limitations in the spatial resolution of light-based approaches need to be considered. These shortcomings have been approached in costly integrated imaging apparatuses offered by several companies, which are unfortunately not readily available for the majority of scientists. Complementing this strategy, SE/BSE provides a powerful tool for surface imaging and simultaneously localising proteins of interest. The presented method expands the existing repertoire of SE/BSE protocols. Building on previous studies [14], we developed a protocol to merge cytosolic protein localisation with surface information in cultured cells. Considering that electron microscopy provides a resolution of at least 1 nm [34], which is one order of magnitude higher than what can be achieved with fluorescence microscopy, this technique overcomes not only problems with alignment but also with spatial resolution.

Antibodies are popular for their use in specifically targeting proteins through unique epitopes in antibody–antigen interaction. Combining this specific localisation to a protein with an immunogold particle, the conjugate displays excellent electron-scattering properties, and has emerged over the use of peroxidase and other enzymatic markers, as it only slightly obscures the structural details of the specimen. Furthermore, the gold signal can be detected and quantified even through biomaterial (20 nm particle: ~240 nm depth), which is not possible to the same extent with enzymatic markers. In theory, the existing and purchasable sizes of gold particles attached to antibodies can vary from 5–40 nm (or even greater). However, while bigger gold particles exist, which may further increase detection depth, its use is not advised, as labelling efficiency is likely to decrease with particle size.

### 4.2. Limitations and Further Improvements of the Method

During fixation, preservation of the ultrastructural integrity, as well as the antibody specificity, is of principal importance. At very high magnifications, we occasionally observe small granules that likely represent fixation or coating artifacts (Figure 3d and Appendix A, bottom). To further improve the presented method, the time between removal of the cell from its physiological environment and fixation could be minimized. This could be accomplished, for instance, by adding fixative right after removing the physiological culture medium. Likewise, to minimize alterations in cell morphology, the fixative could be buffered (0.1–0.2 M). Alternatively, initial chemical fixation could be replaced by instantaneous cryo-fixation, high-pressure freezing, or cryo-SEM, which would prevent cell shape rearrangements during fixation [35].

Another concern with sample quality during unroofing is the lack of control and the substantial mechanical sheer forces exerted to the fixed sample. Here, cryo-fracture or focused ion beam milling combined with scanning electron microscopy would yield a more elegant approach to expose the cell interior.

Based on our findings (Figure 2a,b), the simultaneous detection of two antigenous structures (e.g., two proteins) in cellular structures via secondary antibodies conjugated to gold particles of different sizes, is readily possible with the presented method. However, its accuracy is only as good as the quality of the primary antibody, which should be carefully tested prior to use. Note that published work indicates that the labelling efficiency of gold-conjugated antibodies is reduced compared to fluorescently tagged antibodies, and therefore the concentration of the labelling substrate needs to be increased to yield reliable results [36].

In our case, and unlike the labelling of proteins on the outer surface of the specimen [37], the back-scattered electrons travel through electro-dense material, which yields substantial scattering (i.e., the “halo”). Conceptually, this halo could be used, analogous to our biomimetic measurements (Figure 2a,b), to determine the size and depth of gold particles within biological samples. However, unlike a uniform carbon layer, the cytosol contains structures that may differ in density (e.g., organelles and the cytoskeleton). Similarly, patterned or curved surface topographies, ubiquitous in biological specimens, are likely to disturb the back-scattered electrons. Hence, to reliably quantify sample depth, statistical comparison to a reference point with a known depth (i.e., relative comparison of proteins) or deconvolution (i.e., absolute depth analysis of proteins) may prove two alternative strategies. In both cases, to accurately determine the position of these structures, these images should be acquired at an appropriate (i.e., high) magnification and pixel resolution. Finally, we would like to note that the presented method relies on conventional antibodies, which are 10–15 nm in size [38,39]. Here, the combination of primary and secondary antibodies yields a distance between the epitope and the gold particle of up to 30 nm. While this lateral inaccuracy is negligible for the presented analysis of curvature-dependent protein recruitment to membrane deformations with a radius of 100–200 nm, the quantification of smaller membrane structures, such as endocytotic sites, may benefit from more accurate fixation and labelling procedures. Although not investigated here, other studies have demonstrated that localisation inaccuracies could be further improved by aptameres or nanobodies [38,39].

## 5. Conclusions

Classical correlative approaches rely on the sequential acquisition of fluorescence-based protein localisation, followed by the scanning electron-based acquisition of the cellular ultrastructure. Considering that image acquisition occurs in two different microscopes and is interrupted by embedding preparations, these time-consuming approaches are subject to substantial alignment errors. By replacing fluorescence-based protein detection with gold-conjugated probes, protein localisation can be investigated together with its morphological aspects in intact cultured cells. The method complements existing SE/BSE protocols for the parallel detection of ultrastructural features and surface proteins, enabling the correlated analysis of plasma membrane topography and cytosolic proteins in cultured cells. This method is, however, not well-suited for the detection of proteins located deep in the cytoplasm (i.e., over 250 nm) or bound to endo-membranes that likely will be damaged by Triton-X 100 [40].

## Figures and Tables

**Figure 1 cells-09-01329-f001:**
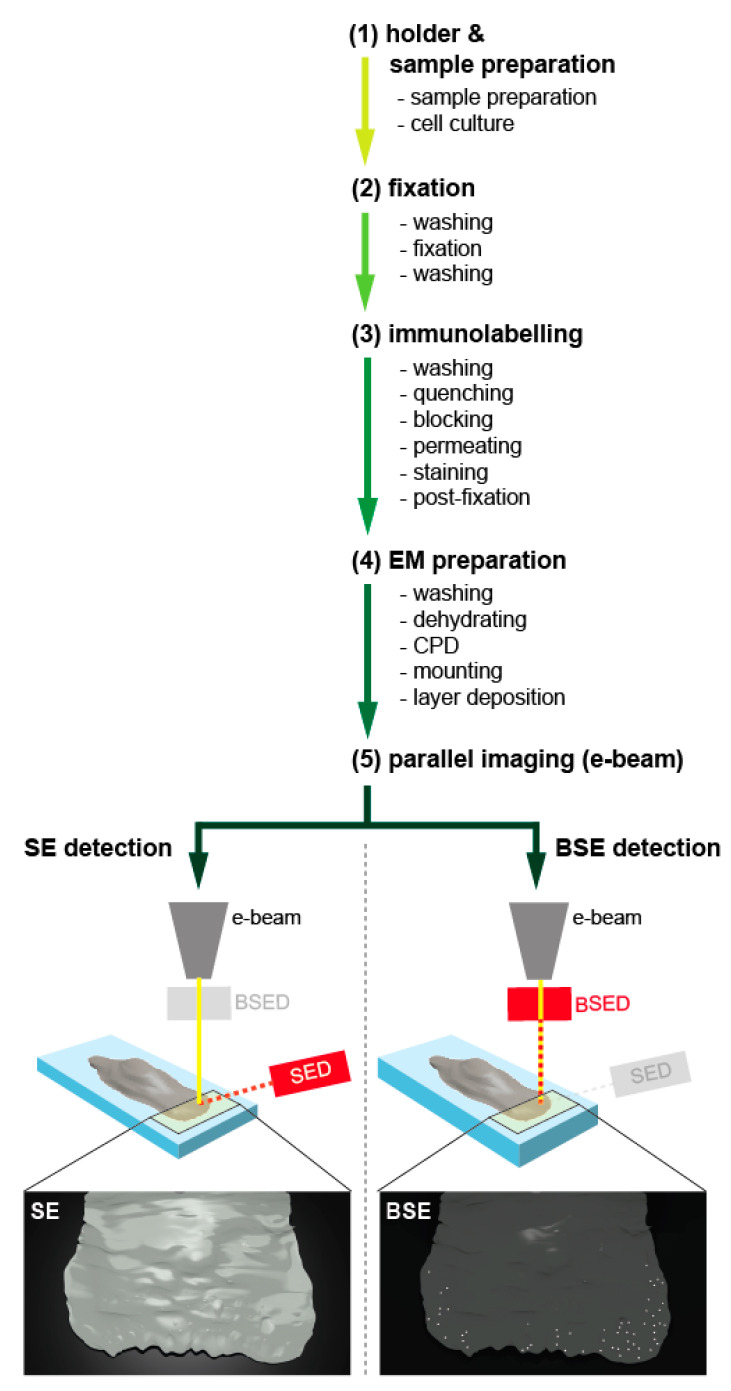
Overview of experimental design. The graphical summary depicts the five main components of the experimental setup. After holder and sample preparation (**1**), which is individually designed according to the requirements of the electron microscope, the biological sample has to be prepared. Pre-fixation ensures structure and protein stability (**2**), and the labelling procedure, which needs to be optimized with respect to the sample, the protein of interest, and the respective antibody, endures the protein localisation (**3**). Before the sample can be imaged, it has to be finalised for EM preparation (**4**). Finally, the probe can be imaged on the same microscope, simultaneously collecting the secondary electrons (SE, **left**) for ultrastructural information and the back-scattered electrons (BSE, **right**) for the visualisation and localisation of the gold-conjugated antibodies (**5**).

**Figure 2 cells-09-01329-f002:**
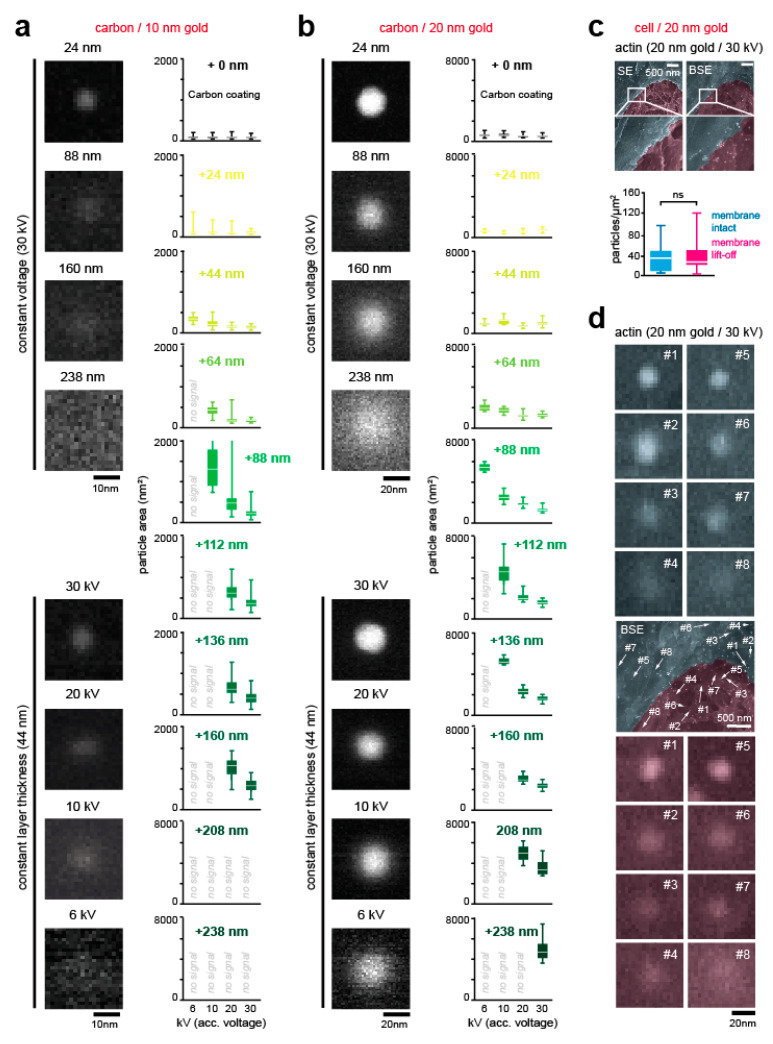
Validation of gold particle detection accuracy. (**a**,**b**) Detection of 10 nm (**a**) and 20 nm (**b**) gold particles through carbon layer via BSE detector. To the left, images of gold particles acquired at 50,000× magnification (1.45 nm per pixel), with an acceleration voltage of 30 kV. Samples were imaged with constant acceleration voltage and varying layer thickness (**top**) as well as with varying acceleration thickness and constant layer thickness (**bottom**). To the right, analysis of the scattering area. From top to bottom, gold particles were deposited on an EM grid (0 nm) and coated with increasingly thick carbon layers. For each thickness, acceleration voltages of 6 kV, 10 kV, 20 kV, and 30 kV were applied. Boxplots depict the scattering area of back-scattered electrons. **(c)** Partially unroofed cell samples. Samples were prepared and labelled with a primary antibody directed against actin, as mentioned above. In some regions, weakly adhesive tape was used to partially unroof the cell (pink, membrane lift-off), and imaged in SE (**top**) and BSE (**bottom**) mode. Images of 20 nm gold particles were acquired at 25,000× magnification (2.89 nm per pixel), with an acceleration voltage of 30 kV. For statistical analysis, gold particles were counted in intact membrane areas (cyan) and compared to unroofed areas (pink) (intact membrane, *n* = 20 images; lift-off membrane, *n* = 11 images; *t*-test with Welch’s correction *p* < 0.5969; Mann-Whitney test *p* = 0.6349). (**d**) BSE of gold-labelled endogenous actin in cellular sample shows spherical blurring. Antibodies directed against actin were labelled with 20 nm gold particles and acquired at 25,000× magnification (2.89 nm per pixel), with an acceleration voltage of 30 kV. Note the “halo” in regions with intact membranes (**top**, cyan) and where membrane was removed (**bottom**, pink). Scale bar: (**a**) 10 nm, (**b**) 20nm, (**c**) 500 nm, and (**d**) 500 nm and 20 nm.

**Figure 3 cells-09-01329-f003:**
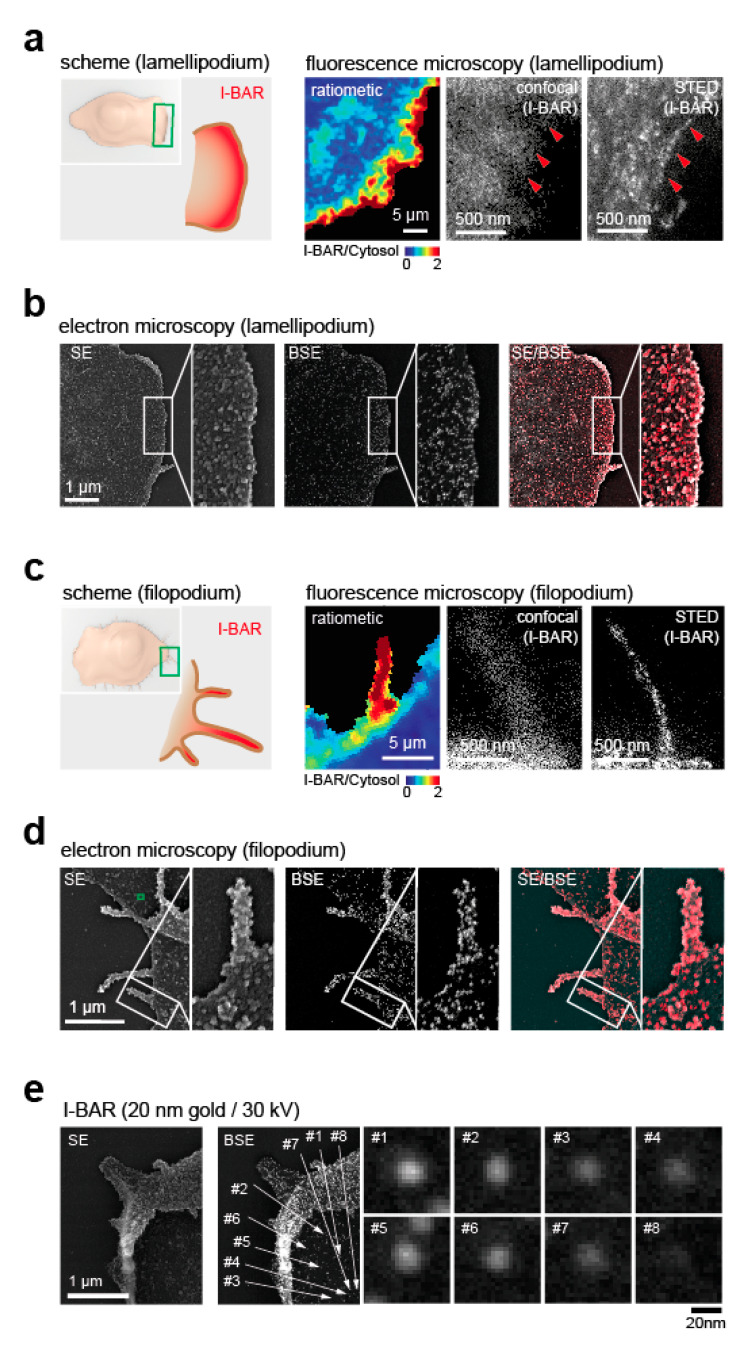
Validation of immunogold particle detection. (**a**) Fluorescence images of lamellipodia. To the left, scheme of I-BAR localisation (red) at the lamellipodial edge. Next to it, ratiometric image of I-BAR vs. cytosolic reference acquired in spinning disc confocal mode (inner **left**), single channel confocal image (inner **right**), and stimulated emission–depletion (STED) image (**right**). (**b**) Correlated SE/BSE of lamellipodium. Samples were transfected with an I-BAR domain tagged to GFP, labelled with a primary antibody against GFP and with a secondary antibody conjugated with 20 nm gold particles. The samples were captured in SE (**left**) and BSE mode (**middle**), and SE/BSE was merged for the region of a lamellipodium (**right**). Both images were acquired at 13,000× magnification (5.57 nm per pixel), with an acceleration voltage of 30 kV. (**c**) Fluorescence images of filopodia. To the left, scheme of curvature-dependent I-BAR enrichment (red) in filopodia. Next to it, confocal images of I-BAR vs. cytosolic marker (inner **left**), single-channel confocal I-BAR (inner **right**), and STED (**right**). (**d**) Correlated SE/BSE of filopodia. Images were captured in SE (**left**) and BSE (**middle**) modes, and SE/BSE were merged (**right**) for the region of a filopodium. Images were acquired at 13,000× magnification (5.57 nm per pixel), with an acceleration voltage of 30 kV. (**e**) BSE of gold-labelled I-BAR shows spherical blurring. Leading edge of cell labelled with antibodies directed against the curvature-sensitive I-BAR domain and imaged in SE (**left**) and BSE (**middle**). To the right, individual gold particles from the cell surface behind the leading edge are shown. All images were acquired at 25,000× magnification (2.89 nm per pixel), with an acceleration voltage of 30 kV. Scale bar: (**a**) 5 µm, 500 nm, (**b**) 1 µm, (**c**) 5 µm, 500 nm, (**d**) 1 µm, and (**e**) 1 µm and 20 nm.

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
