# Peer review of "Parallel Acquisition of Plasma Membrane Ultrastructure and Cytosolic Protein Localisation in Cultured Cells via Correlated Immunogold SEM"

_cells, 2020, doi:10.3390/cells9061329_

Round 1
Reviewer 1 Report
The manuscript describes a method for the localization of proteins on, or close to, the cell surface, together with cell surface topography, using scanning electron microscopy.
There are some aspects in the manuscript that need to attention before publication can be considered.
- It is not recommendable to wash the cells in PBS before fixation. PBS (with or without addition of sucrose) is not a physiological medium. Thus, it is likely to cause alterations in cell morphology and protein localization. It is much better to add the fixative right after removing the physiological culture medium, without any washing. In addition, the fixative should be properly buffered. A concentrated buffer, 0.1 to 0.2 M, is recommended for electron microscopy, instead of PBS.
- Figure 1 is not optimal. It is not clear enough that the grey slab with brown spots on it represents a cover slip with cells attached to it. It is also likely that the samples were not incubated in Erlenmeyr flasks at any point. Photos of different sample preparation steps might be more informative and less misleading.
- It is widely known that 10-nm and 20-nm gold particle do not penetrate into cells, if the ultrastructure of the cytoplasm is intact. It is also known that Triton-X 100 is not compatible with TEM immuno electron microscopy since it destroys membrane structure (instead, milder detergents such as saponin or digitonin are used). Since the authors used TritonX-100 twice for permeabilization, it seem obvious that the current protocol is suitable for localization of proteins that locate on the plasma membrane, or immediately below the plasma membrane, preferably on the cytoskeleton. However, it is also obvious that the protocol is not suitable for localization of proteins locating deeper in the cytoplasm, on membrane-bound organelles, or in the nucleus. This should be stated more clearly in the manuscript.
- There is a problem concerning the positive and negative controls for immunostaining, presented in Figure 2C and Supplement Figure 1b. Why does the SE image look different in negative and positive controls? It seems that the positive control lacks the plasma membrane, while the plasma membrane is intact in the negative control.
Author Response
A detailed point-by-point rebuttal can be found in the attached PDF file.

Reviewer 2 Report
The authors propose a new technique to identify protein localization in the depth of few nm of cells by combining secondary and backscattered electron detection.
Although conceptually I find the approach interesting I consider that the results presented by the authors do not demonstrate the proposed ideas:
Major points:
Results:
From the concept of the manuscript, the results should be able to demonstrate the location of gold particles at different depths associated to different proteins with good ultrastructural preservation. However, the images do not allow the identification of these features.
- Fig 3d. SEM images. Immunogold experiments claim that "I-BAR could be reliably aligned with membrane ultrastructure within individual filopodia"… However, the ultrastructure of the filopodia is not well preserved and even less the membrane ultrastructure. The biological membranes are not well preserved and rather composed of small granules, which could represent coating artifacts or fixation artifacts. The authors only fix with formaldehyde and this weak fixation could cause some problems when preparing the sample and affect the preservation of the ultrastructure. They do not show a standard preparation for membranes to compare with their results, such as high-pressure freezing, freeze-fracturing, or even better cryo-SEM). The last would be the absolute gold standard. Experiments to improve the sample quality would be necessary.
- Immunogold particles seem to be everywhere. The method claims to identify proteins at deep locations but, no clear experiment demonstrate this.
- In this line, the author demonstrate that is possible to identify gold particles in the depth of carbon layers by different acceleration voltages. However, they do not discuss how to differentiate between gold particles located at different highs in the sample, which could be more realistic in a biological sample. The value of the "halo" could be used to determine the depth of a particle associated to the acceleration voltage. This could be used as a calibration for the particles at different depth positions, providing an interesting approach to the method.
- To demonstrate the depth location of gold-conjugated antibodies FIB/SEM experiments are more accurate compared to the "unroof" approach.
- They should also consider and mention the work of De Goede, M., Johlin, E., Sciacca, B., Boughorbel, F., & Garnett, E. C. (2017). 3D multi-energy deconvolution electron microscopy. Nanoscale, 9(2), 684-689.
- In the introduction, the resolution of electron microscopes is mentioned of smaller than 1nm. However, in the result/discussion part nothing is said about the resolution achieved in the present method. The results should show the resolution achieved in terms of membrane details as well as depth information.
- In this line, in discussion: “Building on previous studies, we developed for the first time a protocol to merge cytosolic protein localisation with surface information in whole-cell samples on the single nanometre scale". The resolution of the method is much larger than was is state here.
- The pixel size of the images is a necessary information to recognize whether the images are acquired at proper sampling criteria.
- The method cannot be applied to whole cells in terms of the complete depth of the whole cells. Thus, it should be corrected and mentioned the specific values achieved with the method. Title, abstract and text should be corrected.
- The confocal and STED images are of poor quality, and do not seem to resolve well the structures. The confocal images in Fig.3a and 3c show too much blur for that technique.
Material and Methods:
the description of the holder is not clear. In the drawings the glass holder looks like contains cavities for the cells. More details and dimensions are necessary to understand the scheme of Fig. 1.
Minor points.
More relevant references for CLEM and super-resolution method are needed.
Figure 3 legend. Scale bar indication is wrong a, b and again a, b instead of c, d.
Author Response

(The authors gave the same response as above.)

Round 2
Reviewer 2 Report
I recognize the effort the authors invested to improve the manuscript which now is clearer and more specific in their results and discussion. However, still I have serious concerns about the results presented.
Ideally new experiments should be performed to improve the quality of the results and the value of the protocol.
Results:
In the new text, in terms of resolution, the authors clearly state that with the current methodological sample preparation the protein of interest (I-BAR) could be determined but, for higher accuracy better sample preservation methods should be used. I think that it should be mentioned in the abstract the resolution that can be obtained with the current protocol. Example: “Collectively, the presented method enables rapid and precise analysis of cytosolic proteins adjacent to cellular membranes with a resolution in the range of 100-200nm”.
Otherwise, the readers could be confused by the resolution of electron microscopes and they have to be aware of the sample preparation limitation that often reduces this high resolution.
In Figure 3e. The halo could be an interesting result but, the ultrastructural information is missing, this method would require of further experiments to provide recognizable structures in the sample. With the current method it could only state that the protein is at certain depth in the cell but, without the morphology it cannot be determined in which organelle the labelling is. Again, the disadvantage of this result should be discussed. I did not find mentioned in the text.
Figure 3d. In this new image, the sample is shown in a resolution where neither the membrane details nor the gold particles are clearly visible. If not better images can be obtained, the previous higher magnification image should be used but, mentioning the limitations in the sample preparation.
I suggest that “whole cells” in the title and in the abstract is exchanged by “cultured cells”. This term is precise and does not imply whether the cells could be imaged in 3D as whole cells could be understood.
Minor point: Figure 3e is mentioned as d.
Author Response
I recognize the effort the authors invested to improve the manuscript which now is clearer and more specific in their results and discussion. However, still I have serious concerns about the results presented. Ideally new experiments should be performed to improve the quality of the results and the value of the protocol.
Reply: We thank the referee for the detailed and constructive critique of the manuscript. To further strengthen our core findings, we added ultrastructural information to the ‘halo’ experiments and correlative images with higher resolution. Following the referees suggestion, we further revised the figures, changed the flow of the story and clarified limitations of the presented model in the text. We hope these changes will help to resolve the remaining concerns.
Results:
(1) In the new text, in terms of resolution, the authors clearly state that with the current methodological sample preparation the protein of interest (I-BAR) could be determined but, for higher accuracy better sample preservation methods should be used. I think that it should be mentioned in the abstract the resolution that can be obtained with the current protocol. Example: “Collectively, the presented method enables rapid and precise analysis of cytosolic proteins adjacent to cellular membranes with a resolution in the range of 100-200nm”. Otherwise, the readers could be confused by the resolution of electron microscopes and they have to be aware of the sample preparation limitation that often reduces this high resolution.
Reply: This is an excellent suggestion. We edited the abstract as follows (lines 17-20): ‘Collectively, the presented method enables rapid and precise analysis of cytosolic proteins adjacent to cellular membranes with a resolution of ~100 nm, without time-consuming preparations, or errors induced by sequential visualisation present in fluorescence-based correlative approaches.’
(2) In Figure 3e. The halo could be an interesting result but, the ultrastructural information is missing, this method would require of further experiments to provide recognizable structures in the sample. With the current method it could only state that the protein is at certain depth in the cell but, without the morphology it cannot be determined in which organelle the labelling is. Again, the disadvantage of this result should be discussed. I did not find mentioned in the text.
Reply: True. In the revised manuscript, we substantially expanded this data set, which now includes ultrastructural information for both staining (see Figs. 2d and 3e).
Disadvantage of the approach are disclosed as follows: (lines 317-326): ‘In our case, and unlike labelling of proteins on the outer surface of the specimen 38, the back-scattered electrons travel through electro-dense material, which yields substantial scattering (i.e. halo). Conceptually, this halo could be used, analogous to our biomimetic measurements (Figs. 2a, b), to determine size and depth of gold particles within biological samples. However, unlike a uniform carbon layer, the cytosol contains structures that may differ in density (e.g. organelles, cytoskeleton). Similarly, patterned or curved surface topographies, ubiquitous in biological specimen, are likely to disturb the back-scattered electrons. Hence, to reliably quantify sample depth, statistical comparison to a reference point with known depth (i.e. relative comparison of proteins) or deconvolution (i.e. absolute depth analysis of proteins) may prove two alternative strategies.’
(3) Figure 3d. In this new image, the sample is shown in a resolution where neither the membrane details nor the gold particles are clearly visible. If not better images can be obtained, the previous higher magnification image should be used but, mentioning the limitations in the sample preparation.
Reply: We agree. To emphasize the membrane details, we included in the revised manuscript images with higher resolution.
Limitations of the approach are mentioned as follows (lines 299-306): ‘At very high magnifications, we occasionally observe small granules that likely represent fixation or coating artifacts (Fig. 3d and Suppl. Fig. 2, bottom). To further improve the presented method, the time between removal of the cell from its physiological environment and fixation could be minimized. This could be accomplished, for instance, by adding fixative right after removing the physiological culture medium. Likewise, to minimize alterations in cell morphology, the fixative could be buffered (0.1-0.2 M). Alternatively, initial chemical fixation could be replaced by instantaneous cryo-fixation, high-pressure freezing, or cryo-SEM, which would prevent cell shape rearrangements during fixation 36.’
(4) I suggest that “whole cells” in the title and in the abstract is exchanged by “cultured cells”. This term is precise and does not imply whether the cells could be imaged in 3D as whole cells could be understood.
Reply: We concur. In the revised manuscript, we exchanged ‘whole cells’ by ‘cultured cells’ throughout the whole text (-> 10 replacements in total).
(5) Minor point: Figure 3e is mentioned as d.
Reply: We thank the referee for pointing out this error, which has been corrected in the revised manuscript.